# A Lower Serum Antioxidant Capacity as a Distinctive Feature for Women with HER2+ Breast Cancer: A Preliminary Study

**DOI:** 10.3390/cancers14235973

**Published:** 2022-12-02

**Authors:** Letícia L. D. Santos, Alinne T. F. Silva, Izabella C. C. Ferreira, Adriele V. Souza, Allisson B. Justino, Donizeti W. Santos, Luiz Ricardo Goulart, Carlos Eduardo Paiva, Foued S. Espíndola, Yara C. P. Maia

**Affiliations:** 1Laboratory of Nanobiotechnology Luiz Ricardo Goulart Filho, Institute of Biotechnology, Federal University of Uberlandia, Uberlandia 38402-022, Brazil; 2Molecular Biology and Nutrition Research Group (BioNut), School of Medicine, Federal University of Uberlandia, Uberlandia 38405-320, Brazil; 3Laboratory of Biochemistry and Molecular Biology, Institute of Biotechnology, Federal University of Uberlandia, Uberlandia 38405-302, Brazil; 4Gynecologic Division, University Hospital, Federal University of Uberlandia, Uberlandia 38405-320, Brazil; 5Palliative Care and Quality of Life Research Group (GPQual), Learning and Research Institute, Barretos Cancer Hospital, Barretos 14784-400, Brazil

**Keywords:** breast cancer, HER2, redox status markers, antioxidant capacity, FRAP, FTIR

## Abstract

**Simple Summary:**

HER2 overexpression in breast tumors serves as a biomarker of aggressive cancer and a poor prognosis. This protein contributes to redox imbalance in the pro-oxidative sense. The study aimed to investigate the infrared spectrum wavenumbers obtained by ATR-FTIR and their relationship with the levels of blood redox status markers. Our results indicate that HER2+ Breast Cancer (BC) cases could be distinguished from HER2− BC and benign breast disease (BBD) cases by their lower serum antioxidant capacity. These unprecedented findings allowed us to assess biochemical changes that occur before clinical signs and allowed us to monitor the tumor. Thus, these data advance knowledge about changes revealed by ATR-FTIR and the lower serum antioxidant capacity, showing our preliminary distinctive feature of these HER2+ molecular subtypes. This research should be extended to a larger sample population.

**Abstract:**

The overexpression of HER2 in breast cancer (BC) can contribute to redox imbalance, which is related to damage and structural modification in many biomolecules. To the best of our knowledge, this is the first study that has investigated the infrared spectrum wavenumbers obtained by ATR-FTIR and their relationship with the levels of redox status markers such as reduced glutathione, superoxide dismutase (SOD), catalase, Ferric Reducing Antioxidant Power (FRAP), and protein carbonyl among women with HER2+ BC, HER2− BC, and benign breast disease (BBD). The study was conducted with 25 women, 17 of whom were diagnosed with BC (6 HER2+ and 11 HER2−) and 8 with BBD. Our results indicate HER2+ BC cases could be distinguished from HER2− BC and BBD cases by their serum’s antioxidant capacity [HER2+ BC vs. HER2− BC (AUC = 0.818; specificity = 81.82%; sensitivity = 66.67%); HER2+ BC vs. BBD (AUC = 0.875; specificity = 75%; sensitivity = 83.33%)]. The changes in biochemical terms that occur in serum as a result of the scarcity of antioxidants are related to a peculiar fingerprint in the infrared spectrum obtained by ATR-FTIR. In the serum of women with BBD, the SOD enzyme level is the highest, and this characteristic allowed us to distinguish them from HER2− BC. Finally, data regarding the serological antioxidant capacity of FRAP and the infrared spectrum by ATR-FTIR will allow us to assess biochemical changes that occur before clinical signs, indicating whether changes in therapy or interventions are necessary.

## 1. Introduction

Breast cancer (BC) is a highly heterogeneous disease. Studied worldwide, it is still a major public health problem and causes many deaths each year, with 3,025,471 (33.8% increase) new cases expected to be diagnosed in 2040 [1]. Molecular subtypes are used to determine the prognosis by understanding the disease, which is largely associated with cell receptors [2,3,4]. One of the cell receptors analyzed is the human epidermal growth factor receptor 2 (HER2), as its overexpression or amplification of its gene in breast tumors serves as a biomarker of aggressive cancer and a poor prognosis, which occurs in around 25% of cases [2,4,5].

HER2 overexpression can contribute to redox imbalance in the pro-oxidative sense, also known as oxidative stress, which is related to damage and structural modification in several cellular components, such as DNA, lipids, and proteins [6,7]. Oxidative stress is considered an important factor in developing various diseases, including neurodegenerative, retinal degenerative, type 2 diabetes, atherosclerosis, pulmonary fibrosis, and cancer [8,9,10,11,12].

Reactive oxygen species (ROS) are mainly comprised of neutral molecules (H_2_O_2_), radicals (hydroxyl radicals), and ions (superoxide), which are produced in greater quantity in BC [6,7]. Cells have natural defense systems against ROS that consist of antioxidants, molecules that inhibit or completely scavenge the action of ROS and protect organisms from oxidative damage [13].

Redox reactions are the basis of the energy-delivering processes in living organisms. The oxidation of a molecule affects its electronic density distribution, altering its vibrational spectrum [14]. The Attenuated Total Reflection-Fourier Transform Infrared (ATR-FTIR) is a vibrational spectroscopy technique that is sensitive to even a 0.002% change in bond strength, reflecting the chemical composition of a sample [13,14]. Thus, this approach can obtain a “biochemical signature of the tumor”, a critical factor for predicting a patient’s response to treatment [15,16].

Our study aimed to investigate the infrared spectrum wavenumbers obtained by ATR-FTIR and their relationship with the levels of redox status markers GSH, SOD, CAT, FRAP, and protein carbonyl among women with HER2+ BC, HER2− BC, and BBD. Our hypothesis is that women with HER2+ BC have a worse redox imbalance. To the best of our knowledge, this is the first study that has investigated infrared spectrum wavenumbers obtained by ATR-FTIR and their relationship to redox status among women with HER2+ BC, HER2− BC, and BBD.

## 2. Materials and Methods

### 2.1. Study Design, Sample Size, and Eligibility Criteria

This study was conducted at a Clinical Hospital in Uberlandia, MG, Brazil, after approval by the Human Research Ethics Committee of the Federal University of Uberlandia (protocol 4.047.065), and all subjects provided written informed consent. The entire study was developed according to the standards of the Declaration of Helsinki and Resolution CNS 466/12.

Women who went to the Clinical Hospital to undergo breast surgery were invited to participate in the study as volunteers. After surgery and histopathological results, the tumors, tissues, and lesions found were classified according to histological type, staging, and status of estrogen receptor (ER), progesterone receptor (PR) and HER2 receptors. HER2 overexpression was identified by 3+ immunohistochemistry or FISH test.

Cases of triple negative BC and cases with suspected or indeterminate cytopathological pattern of malignancy were excluded from the study. Figure 1 shows the number of women screened and enrolled during the study.

A posteriori power analysis was carried out using the G*Power software, version 3.1 (Heinrich-Heine-University Düsseldorf, Germany). The analysis was based on a F-test linear multiple regression with an effect size f of 0.35, an alpha level of 0.15, and 1 tested predictor. Considering a total of 25 participants, the output parameter showed a power of 92% (post hoc: computer achieved power calculation) [17].

### 2.2. Data and Biological Material Collections

The data of the study were obtained from interviews and medical records.

With the volunteer’s consent, 5 mL of peripheral blood was collected to obtain the infrared spectrum by ATR-FTIR and analysis of redox status biomarkers. A vacuum collection tube containing separating gel with a clot activator was used on this occasion, and the samples were processed as indicated by the manufacturer (KASVI^®^, São José dos Pinhais, PR, Brazil).

### 2.3. Infrared Spectrum Evaluation of Serum Samples by ATR-FTIR

Infrared spectroscopy was conducted with the ATR accessory Cary 630 (Agilent, Santa Clara, CA, USA) using MicroLab (Agilent, Santa Clara, CA, USA) software version 5.7, an ATR micro component, crystal material in the ATR unit, and a platinum diamond. The samples were applied in aluminum pellets (10 µL) and heated to 80 °C in a dry bath for 3 min. The spectra were analyzed in the wavenumber region from 650 cm^−1^ to 4000 cm^−1^, with a resolution of 4 cm^−1^.

### 2.4. Determination of Serum Redox Status Biomarkers

For the GSH (U/mg protein) quantification, serum samples were added to metaphosphoric acid (ratio 1:1) and the analysis was conducted as previously reported [18].

SOD (U/mg protein) activity analysis was conducted as previously reported [19].

CAT (U/mg protein) activity was determined based on monitoring the decomposition of H_2_O_2_ at 240 nm in a reaction medium containing 20 mM H_2_O_2_, 10 mM phosphate buffer (pH 7.0), and 10 µL of the sample. The analysis was conducted as previously reported [20].

The serological non-enzymatic antioxidant capacity (µmol/L Trolox) was evaluated by the FRAP method. The antioxidants present in the samples reduce Fe^3+^ to Fe^2+^, which is chelated by 2,4,6-tris(2-pyridyl)-s-triazine (TPTZ) to form an Fe^2+^-TPTZ complex, which has an intense blue color. The analysis was conducted as previously reported [21].

To detect protein carbonyls (nmol/mg ptn) in serum, the samples were incubated with 10 mM DNPH in 2.5 N hydrochloric acid (HCl) in the dark for 1 h and then 20% trichloroacetic acid (TCA) was added. After two centrifugations at 9000× *g* for 5 min, the pellet was resuspended in 6 M guanidine solubilized in 2.5 N HCl and heated at 37 °C for 5 min. The product formed was visualized at 370 nm [22].

### 2.5. Statistical Analysis

Statistical analyzes were conducted using the SPSS Program version 21.0 (IBM, New York, NY, USA) and Prism Program version 9.0 (Graphpad, San Diego, CA, USA). The values of the second derivative of the infrared spectrum were used for statistical analyses. Normality tests were performed for the variables using Shapiro-Wilk. The T-test and the ANOVA test were chosen to search for the redox status biomarkers profile. To determine whether the presence and concentration of the redox status biomarkers may impact the infrared spectrum of sera, statistical analyses using the Generalized Linear Models (GLzM) test were performed. Finally, Spearman’s correlation was also performed. The 95% confidence interval and *p*-value < 0.05 were determined for all tests.

## 3. Results

### 3.1. Characterization of the Study Population

The study was conducted with 25 women, 17 of whom were diagnosed with BC and 8 with BBD.

Table 1 presents the clinical and histopathological characteristics of the study participants with BC. Six are HER2+ and 11 HER2−, with a mean age ± standard deviation (SD) of 63 ± 4.3 years and 54.18 ± 14.4 years, respectively. Regarding the molecular subtype, most women belong to the group with the best prognosis in both cases: luminal B HER2+ (66.6%; *n* = 4) and luminal A (81.8%; *n* = 9).

The volunteers included in the BBD group (*n* = 8) were diagnosed with the following conditions: ductal ectasia, fibrosis, sclerosing adenosis, ductal hyperplasia, fibroadenoma, microcalcification, and giant juvenile fibroadenoma. The average age was 46.88 years, with a minimum of 19 and a maximum of 71 (SD = 14.71).

### 3.2. Wavenumbers Absorbance Related to Oxi-Reduction Differ between Groups Studied by ATR-FTIR

From the averages of the original infrared spectrum of the whole serum from the women with HER2+ BC, HER2− BC and BBD, graphs of the second derivative were obtained, which are shown in Figure 2. With this approach, the influence of scattering is reduced, and there is an information gain.

The main wavenumbers detected in the serum, i.e., those with a difference between groups, as well as their vibrational modes and their molecular sources, are described in Table 2. In the region, 706–710 cm^−1^ OH (hydroxyl group) out-of-plane bending, CH (hydrocarbon chain) out-of-plane bending and NH (amino group) bending (Amide IV) are found [22,23,24]. Wavenumbers 3874 cm^−1^, 3876 cm^−1^, 3878 cm^−1^, 3880 cm^−1^, 3882 cm^−1^, 3887 cm^−1^, 3889 cm^−1^ and 3893 cm^−1^ are part of a region of the infrared spectrum where the stretching vibrational bands of OH (hydroxyl group) and NH (amino group) stretchings are found [22,25,26].

Considering that differences were observed only between the HER2+ BC and BBD spectra, by means of the ROC curve analysis, we searched for a cutoff point so that it is possible to discriminate serum samples from individuals with HER2+ BC with high accuracy.

The only wavenumbers with *p*-value < 0.05 for the ROC curve were 3874 cm^−1^, 3876 cm^−1^ and 3878 cm^−1^, which are shown in Figure 3. All three have high sensitivity and specificity.

### 3.3. Differentiation of Sera Based on Redox Status

Based on the molecular information obtained from the ATR-FTIR spectrum with serum samples, we investigated which redox status marker could better distinguish the sera, initially between two categories: BC and BBD. In this comparison, GSH (*p* = 0.024) and SOD (*p* = 0.002) levels were significantly reduced in women with BC (Table 3).

Afterwards, redox status markers were profiled among women with HER2+, HER2− and BBD (Table 4). The average GSH level was lower in the HER2+ BC group, but with a *p* value = 0.059 (borderline). SOD levels remained higher in the BBD group compared with the others (*p* = 0.010). Post-hoc revealed that the level of SOD contained in serum samples from women with HER2− BC was significantly lower than the level for BBD (*p* = 0.008).

Furthermore, FRAP values were significantly reduced in women with HER2+ BC (*p* = 0.049). Between HER2+ and HER2−, post-hoc presented a *p-value* (*p* = 0.045), thus being the only redox status marker studied able to differentiate these two groups (Table 4).

The figures with the individual levels of GSH, SOD and FRAP in each studied group are presented in Figure 4. In this format, it is possible to observe the homogeneity of the data in each group.

In order to obtain a cut-off point at which it is possible to discriminate the studied groups based on their redox status, the ROC curve analysis was performed. In this step, Figure 5 presents the analyzes with *p* < 0.05. For FRAP a cut-off point of <117 was found to discriminate women with HER2+ BC from women with BBD (specificity 75% and sensitivity 83.33%). Furthermore, this same antioxidant capacity marker also discriminates against women with HER2+ BC from those with HER2− BC, with a cut-off point < 97.96 (specificity of 81.82% and sensitivity of 66.67%).

The SOD enzyme level presented a cut-off point <0.2238 from which discriminates the HER2− BC group from the BBD group (100% specificity and 90% sensitivity).

### 3.4. FRAP Strongly Correlates and Affects Results Obtained by ATR-FTIR in Serum Samples from Women with HER2+ BC

Based on the previous results, we proceeded to investigate whether the absorbance relative to the wavenumbers in the ATR-FTIR that are different between the HER2+ and BBD groups (3874 cm^−1^, 3876 cm^−1^, and 3878 cm^−1^) are influenced by redox status markers GSH, SOD, CAT, FRAP, and carbonylated protein.

Among the redox status markers evaluated, only FRAP impacts the absorbance value obtained at wavenumbers 3874 cm^−1^ (β = −0.038; *p* < 0.001), 3876 cm^−1^ (β = −0.059; *p* < 0.001) and 3878 cm^−1^ (β = −0.066; *p* < 0.001) of the HER2+ BC infrared spectrum. Regarding the BBD group, the SOD level affected the absorbance value obtained at wavenumbers 3874 cm^−1^ (β = 13.525; *p* = 0.034), 3876 cm^−1^ (β = 18.571; *p* = 0.001) and 3878 cm^−1^ (β = 24.565; *p* = 0.042). No markers studied showed influence on the ATR-FTIR infrared spectrum obtained from serum samples from women with HER2− or the general BC group.

To determine whether there is a correlation between the redox status markers and the absorbance values of the selected wavenumbers, statistical analyzes were performed. Using Spearman’s correlation, a correlation between FRAP level and the regions of the spectrum 3874 cm^−1^ (r = 0.829; *p* = 0.042), 3876 cm^−1^ (r = 0.829; *p* = 0.042), and 3878 cm^−1^ (r = 0.829; *p* = 0.042) was found for the HER2+ BC group.

However, for the BBD group, no such correlations occur for FRAP [3874 cm^−1^ (r = 0.143; *p* = 0.736), 3876 cm^−1^ (r = 0.190; *p* = 0.651) and 3878 cm^−1^ (r = 0.119; *p* = 0.779)] or for the other redox status markers.

## 4. Discussion

The present study aimed to investigate the infrared spectrum wavenumbers obtained by ATR-FTIR and their relationship with the levels of redox status markers GSH, SOD, CAT, FRAP and protein carbonyl among women with HER2+ BC, HER2− BC, and BBD. Our results indicate HER2+ BC cases could be distinguished from HER2− BC and BBD cases by their serum’s antioxidant capacity [HER2+ BC vs. HER2− BC (AUC = 0.818; specificity = 81.82%; sensitivity = 66.67%); HER2+ BC vs. BBD (AUC = 0.875; specificity = 75%; sensitivity = 83.33%)]. Furthermore, the changes in biochemical terms that occur in serum as a result of the scarcity of antioxidants, are related to a peculiar fingerprint in the infrared spectrum obtained by ATR-FTIR. These results confirm our initial hypothesis that a worse redox imbalance occurs in HER2+ BC cases. 

Differences were found between the spectra of HER2+ and BBD serum in the wavenumbers 706–710 cm^−1^; 3874 cm^−1^; 3876 cm^−1^; 3878 cm^−1^; 3880 cm^−1^; 3882 cm^−1^; 3887 cm^−1^; 3889 cm^−1^ and 3893 cm^−1^, among which three had ROC curve results with *p* < 0.05 [3874 cm^−1^ (AUC = 0.833), 3876 cm^−1^ (AUC = 0.843) and 3878 cm^−1^ (AUC = 0.843)]. In the serum of HER2+ BC women, the antioxidant capacity is the lowest and impacts the absorbance in wavenumbers 3874 cm^−1^ (β = −0.038; *p* < 0.001), 3876 cm^−1^ (β = −0.059; *p* < 0.001) and 3878 cm^−1^ (β = −0.066; *p* < 0.001). Meanwhile, in the serum of women with BBD, the SOD enzyme level is the highest, which affects the absorbance on the same wavenumbers [3874 cm^−1^ (β = 13.525; *p* = 0.034), 3876 cm^−1^ (β = 18.571; *p* = 0.001) and 3878 cm^−1^ (β = 24.565; *p* = 0.042)], and this characteristic allows us to distinguish them from HER2− BC samples (AUC = 0.92). In addition, the antioxidant capacity, obtained by the FRAP assay, allowed us to distinguish HER2+ BC samples from the others (AUC from HER2+ vs. HER2− = 0.81; AUC from HER2+ vs. BBD = 0.87), and correlated with the wavenumbers 3874 cm^−1^ (r = 0.829; *p* = 0.042), 3876 cm^−1^ (r = 0.829; *p* = 0.042) and 3878 cm^−1^ (r = 0.829; *p* = 0.042).

A pilot study conducted by Kepesidis and contributors (2021), applying FTIR in combination with machine learning for BC detection performed in Saudi Arabia, included 26 women with BC and 26 non-symptomatic women. They concluded from the data obtained with plasma from the volunteers that their model achieved an average performance of 0.79 in terms of area under the curve (AUC) of the receiver operating characteristic (ROC) and found a relationship between the size of the effect of the measured fingerprints and the progression of the tumor [28]. The ROC curve analyzes performed in our study showed AUC values between 0.81 and 0.92, with high sensitivity and specificity. Despite the smaller sample size of our study, the study groups were much more homogeneous, with well-defined characteristics and precise diagnoses, thus obtaining representative results.

Oxidative stress can cause spectral changes due to alterations in intramolecular and intermolecular interactions of the biomolecules present in the blood. The redox status interferes with the conformation of proteins such as human serum albumin [29]. The region of the spectrum called Amide IV (about 620–767 cm^−1^), in which band 706–710 cm^−1^ is found, is sensitive to protein conformation [24]. In this sense, the difference between the absorbances of the HER2+ and BBD groups (Table 2) confirms that there are changes in protein folding in women’s serum with HER2+ BC.

Elmi and colleagues (2017) evaluated serum from healthy women and women with BC using FTIR spectroscopy coupled with principal component and linear discriminate (PCA-LDA) analyses. Differences were found between groups in the regions of the spectrum referring to sugars, proteins, and lipids. It was highlighted that the infrared spectrum region of 3090–3700 cm^−1^ (NH stretching mainly) was the most relevant criterion to distinguish the analyzed groups. The authors said that this fact could be explained by the protein changes that occur during BC, but they do not indicate what these changes could be [30]. In agreement with other studies, the alterations caused by the more accentuated redox imbalance in our HER+ BC serum samples would be due to variations in proteins due to oxidation, leading to modifications of the thiol group and possible alteration of the carbonyl group [28,31].

Currently, to assess the prognosis and predict the therapeutic response of patients with BC, knowledge of several pathological parameters is necessary. The combination of these parameters brings great clinical value in regard to medicine and nutrition, as it is possible to classify patients into many risk categories [32]. Among the 17 women with BC who participated in the present study, only two (11.76%) were classified as G1 (well-differentiated tumor), 11 (64.70%) as G2 (moderately differentiated tumor) and four (23.52%) as G3 (poorly differentiated tumor). Analyzing the HER2+ and HER2− groups separately, it can be observed that tumors with a higher histological grade are concentrated in the HER2+ group (all G3 cases). Poorly differentiated tumors are considered to have a poor prognosis because they have a cell phenotype similar to that of stem cells, becoming more aggressive and with greater metastatic potential [33]. Thus, considering all the histopathological characteristics described in Table 1, the HER2+ BC group is notably the one with the worst prognosis among the study groups.

Based on this information, it is possible to affirm that our results indicate a lower serological antioxidant capacity among women with a poor prognosis for BC. However, it is not known whether this would be the cause or consequence of this clinical condition. Nevertheless, there are studies that associated low antioxidant capacity with increased risk of mortality from all causes, including cancer and cardiovascular disease [33,34,35,36,37].

It is well known that diet is a key modifiable risk factor for several diseases, including BC. In the present study design, we did not collect data on the volunteers’ diet, but our research group previously published a prospective study carried out among women with BC undergoing chemotherapy, where we found a decrease in the total antioxidant capacity of their diets over the time of treatment and among women that presented post-treatment metastasis [38].

Chemotherapy drugs, such as doxorubicin, induce cell death by producing ROS [39]. Consequently, this leads to the assumption that antioxidants could disrupt the treatment. However, more studies need to be conducted in order to establish a value from which the serum’s antioxidant capacity would have a real effect on the non-apoptosis of tumor cells. In this sense, the ATR-FTIR and FRAP methods can contribute to the advancement of this area of knowledge, providing answers to these questions.

GSH is considered the most abundant molecule among endogenous antioxidants. It can eliminate ROS directly or indirectly, reacting with superoxide radicals and some other ROS, produced by xenobiotic metabolism and inflammatory responses, or revitalizing other antioxidants, such as vitamin E [40]. In our study, GSH levels in women with BC were lower when compared with women with BBD (Table 3), indicating a low defense potential against oxidative damage in the BC group. In addition, a research group pointed out that an increase in nitric oxide synthesis occurs as a consequence when GSH levels decrease in the body, as there is oxidative activation of the main transcription factor related to inflammation, called NFκB, and also an increase in the enzymatic activity of induced nitric oxide synthase (iNOS), features that are closely related to oxidative stress [41].

The SOD enzyme levels found in the serum of women with BC in general were lower when compared with women in the BBD group in this study (Table 3). SOD provides cellular defense against the superoxide radical. Early work with SOD in cancer already observed a lower level of this enzyme in the types of tumors analyzed [42]. Currently, it is known that these low levels of SOD only occur during tumor progression due to inactivation by proteins c-jun and p53 [6,43,44,45]. The scarcity of SOD can lead to an increase in ROS and consequently significant accumulation of mutations.

Indeed, in advanced stages, SOD levels rise in response to the excessive increase in ROS and p53 mutation [6,44,45]. In our sample, none of the volunteers with HER2− BC have grade 3 tumors and only 18.1% were classified in clinical stage III, while 66.6% and 50% of the HER2+ group have these features, respectively (Table 1). This may be one of the reasons for the HER2− BC group to present a lower level of SOD than the others (Table 4). 

Regarding CAT, it is considered the main antioxidant enzyme, present mainly in peroxisomes of mammalian cells [13]. There are studies linking its lower activity to tumor metastasis [46,47,48], while others point to its overexpression in malignant cells, as it combats the increase in H_2_O_2_ generated in large amounts in cancer cells and thus helps the survival of the malignant cells [49,50]. Chromatin remodeling is the main regulatory event of CAT expression during the process of acquisition of resistance by malignant cells in BC against oxidative stress [51]. Our study did not find different levels of CAT between the groups analyzed, but there is a tendency for cases of BC to have a higher average level of this enzyme (Table 3). 

Protein carbonylation is a trace generated by oxidative damage and a way to measure its extent in body fluids and tissues [52,53]. When produced, the carbonyl groups are chemically stable, which facilitates their detection. Physiological levels are low, around 1 nmol/mg of protein, but during oxidative stress, its concentration may increase 2 to 8 times [54]. According to our results, the average level of carbonyl protein was above 1 nmol/mg of protein in the three groups analyzed (Table 3 and Table 4). However, even with a greater amount in the HER2+ BC group (average = 5.629 nmol/mg of protein), the difference was not significant between the groups for this biomarker.

The prognosis for HER2+ BC has improved over the years with the development of specific drugs targeting this receptor, both monoclonal antibodies (trastuzumab and pertuzumab), as well as tyrosine kinase inhibitors (lapatinib, tucatinib) and drug-conjugated antibodies (ado-trastuzumab emtansine and trastuzumab deruxtecan), which are used in combination with chemotherapy or alone in different clinical scenarios [55]. Though, in many cases, selective therapeutic pressure leads to more aggressive tumor recurrence, even after a long period of treatment efficacy [56]. Thus, it is of paramount importance that more research is done to be able to predict and monitor the response to early treatment. 

Although there are studies investigating the use of FTIR for cancer diagnosis [16,57,58,59,60,61,62], this was not the objective of the present study. Here we pioneered combining infrared spectrum analyzes and levels of redox status markers in the blood, expanding the frontier of scientific knowledge in this area for women with HER2+ BC, HER2− BC, and BBD. We emphasize that, as a preliminary study, the results are reliable according to the a posteriori sample calculation, although the data presented are still from a small number of women. Another possible limitation is that, in addition to cancer, there are other comorbidities in the studied population. However, no differences were found for the values of the redox state markers when dichotomized into with and without comorbidities (results not shown).

For a new approach to be applicable in clinical practice, accuracy is important, but it is not enough. Due to this, even though there is development in this area, access is still restricted. A wide range of sample types can be tested in the FRAP assay and can be performed manually, semi-automatically, or even adapted to run in fully automated mode using a program in a biochemical analyzer [63]. Thus, the quantification of the serum’s antioxidant capacity can be accessed quickly, cheaply, and safely and inserted in the practice of clinical analysis laboratories.

Studies carried out with BC patients had already reported a lower FRAP level in disease samples when compared with healthy ones [34,64], but there is no precedent showing HER2+ BC cases to be distinguished from HER2− BC cases by their serum’s antioxidant capacity (reducing power). Thus, our preliminary results are of great value to the scientific community as they indicate a distinctive feature of these HER2+ molecular subtypes, which should be extended to a larger sample population. More studies should be conducted with this innovative approach combining the ATR-FTIR technique and the evaluation of the redox status of women with BC.

## 5. Conclusions

The present study concludes, among the redox status biomarkers evaluated, that the lowest antioxidant capacity is an important characteristic in the serum of women with HER2+, distinguishing them from other groups. The changes in biochemical terms that occur in serum as a result of the scarcity of antioxidants are related to a unique fingerprint in the infrared spectrum obtained by ATR-FTIR. Finally, data regarding the serological antioxidant capacity of FRAP and the infrared spectrum by ATR-FTIR will allow us to assess biochemical changes that occur before clinical signs, indicating whether changes in therapy or interventions are necessary throughout cancer treatment.

## Figures and Tables

**Figure 1 cancers-14-05973-f001:**
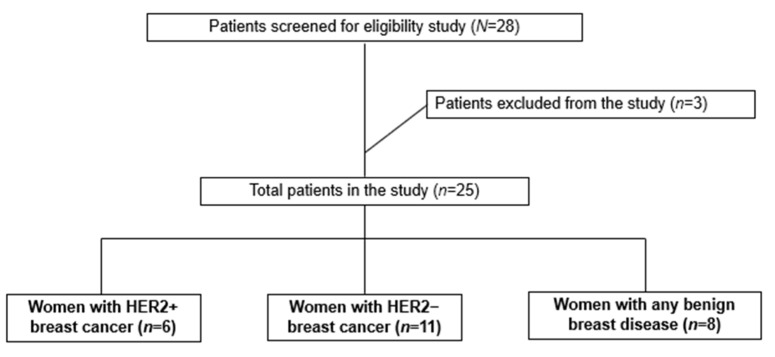
Study flowchart.

**Figure 2 cancers-14-05973-f002:**
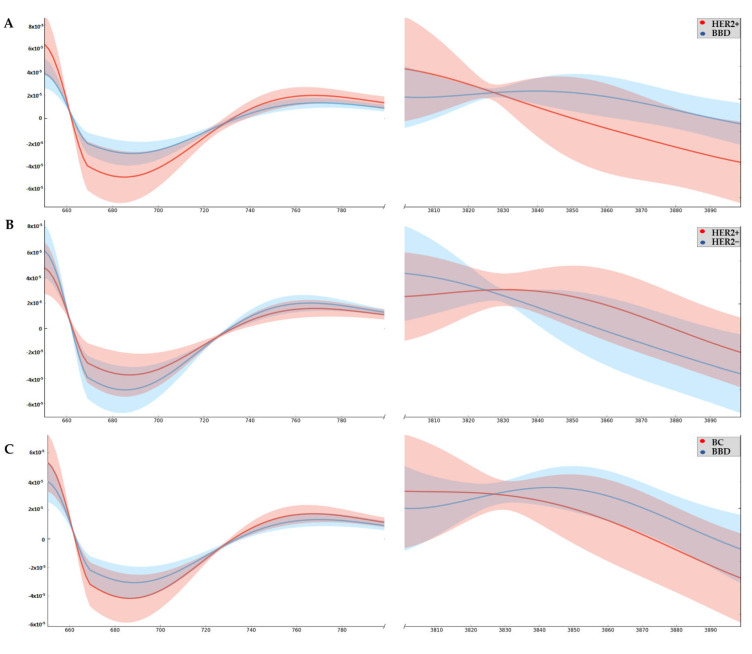
Graph of the second derivative ATR-FTIR. (**A**) Infrared spectrum HER2+ vs. BBD; (**B**) Infrared spectrum HER2+ vs. HER2−; (**C**) Infrared spectrum BBD vs. BC.

**Figure 3 cancers-14-05973-f003:**
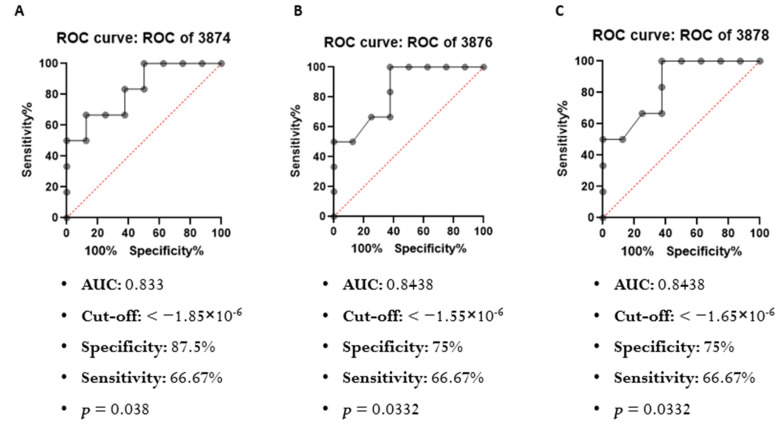
ROC Curve for HER2+ vs. BBD. (**A**) 3874 cm^−1^; (**B**) 3876 cm^−1^; (**C**) 3878 cm^−1^.

**Figure 4 cancers-14-05973-f004:**
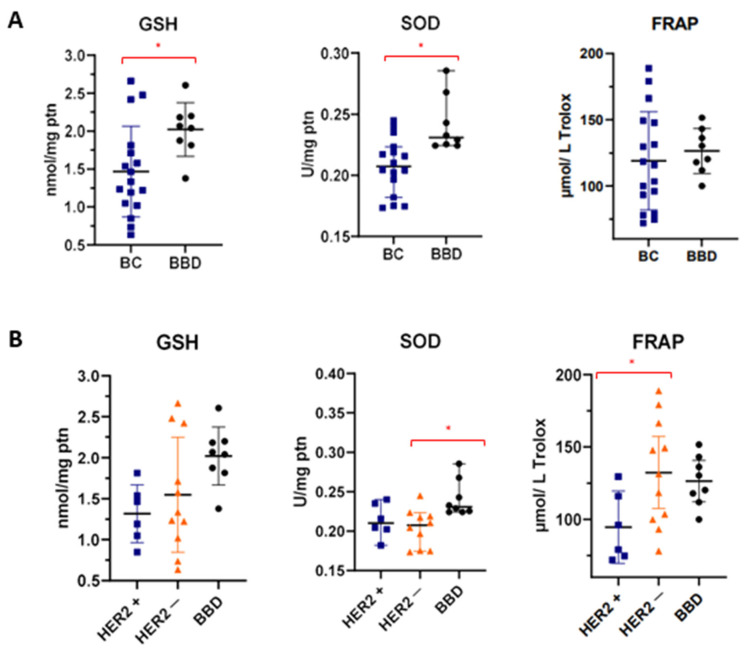
Redox status biomarkers levels in BC, HER2+, HER2− and BBD women. (**A**) GSH, SOD and FRAP levels in BC and BBD women; (**B**) GSH, SOD and FRAP levels in HER2+, HER2− and BBD women. *, denotes statistical difference between the two highlighted groups.

**Figure 5 cancers-14-05973-f005:**
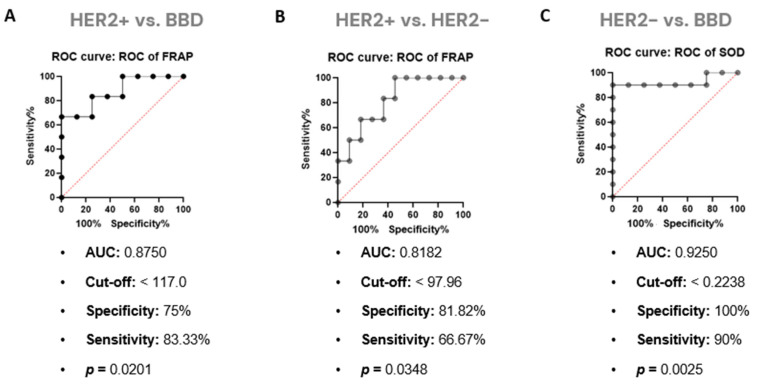
ROC Curve for HER2+ vs. BBD, HER2+ vs. HER2− and HER2− vs. BBD. (**A**) FRAP differentiates the groups HER2+ and BBD; (**B**) FRAP differentiates the groups HER2+ and HER2−; (**C**) SOD differentiates the groups HER2− and BBD.

**Table 1 cancers-14-05973-t001:** Clinical and histopathological characteristics of women with breast cancer (*n* = 17).

Characteristics	HER2+ *n* (%)	HER2− *n* (%)
Age (years) mean (min–max ± SD)	63 (58–71 ± 4.3)	54.18 (29–75 ± 14.4)
Menopause		
No	-	4 (36.3)
Yes	6 (100)	7 (63.6)
Tumor Subtype		
Ductal carcinoma	5 (83.3)	9 (81.8)
Lobular carcinoma	1 (16.6)	2 (18.1)
Clinical Stage		
I	2 (33.3)	3 (27.2)
II	1 (16.6)	5 (45.4)
III	3 (50)	2 (18.1)
NR	-	1 (9)
Histological Grade		
G1	-	2 (18.1)
G2	2 (33.3)	9 (81.8)
G3	4 (66.6)	-
Molecular Subtypes		
ER+ and/or PR+, HER2− and Ki-67 < 14%	-	9 (81.8)
ER+ and/or PR+, HER2− and Ki-67 ≥ 14%	-	2 (18.1)
ER+ and/or PR+, HER2+	4 (66.6)	-
ER−, PR− and HER2+	2 (33.3)	-
Surgery		
Radical Mastectomy	1 (16.6)	4 (36.3)
Conservative Surgery	5 (83.3)	6 (54.5)
Others	-	1 (9)

HER2+, human epidermal growth factor receptor 2 positive group (*n* = 6); HER2−, human epidermal growth factor receptor 2 negative group (*n* = 11); SD, standard deviation; G1, well-differentiated tumor (low grade); G2, moderately differentiated tumor (intermediate grade); G3, poorly differentiated tumor (high grade); ER, estrogen receptor; PR, progesterone receptor; −, negative; + positive; Ki-67, Ki-67 antigen; NR, not registered.

**Table 2 cancers-14-05973-t002:** Assignments and comparison between second derivative serological ATR-FTIR spectra of women with HER2+ BC (*n* = 6), HER2− BC (*n* = 11) and women with BBD (*n* = 8).

2nd Derivative Wavenumber/cm^−1^	HER2+ vs. BBD *p*	HER2+ vs. HER2− *p*	BC vs. BBD *p*	Proposed Vibrational Mode	Molecular Source
706–710	0.049	0.250	0.122	OH out-of-plane bending; CH out-of-plane bending; NH bending	Hydroxyl group; Amide IV
3874	0.042	0.207	0.125	OH stretching; NH stretching	Hydroxyl group; Amino group
3876	0.032	0.218	0.122		
3878	0.032	0.219	0.015		
3880	0.048	0.223	0.111		
3882	0.048	0.227	0.115		
3887	0.044	0.247	0.117		
3889	0.045	0.247	0.120		
3893	0.047	0.275	0.122		

Assignments based on different references [23,24,25,26,27]. T-test analysis of the second derivative spectra of women with HER2+ BC, HER2− BC and women with BBD.

**Table 3 cancers-14-05973-t003:** Profile of redox status markers among women with breast cancer and benign breast disease at a university hospital in Uberlandia, Minas Gerais, Brazil (*n* = 25).

Markers	BC (*n* = 17)		BBD (*n* = 8)		*p*
	Mean ± SD	Median (p25–p75)	Mean ± SD	Median (p25–p75)	
GSH	1.467 ± 0.599	1.335 (1.034–1.763)	2.021 ± 0.354	2.054 (1.831–2.195)	0.024 ^θ^
SOD	0.207 ± 0.023	0.207 (0.185–0.222)	0.241 ± 0.023	0.230 (0.224–0.261)	0.002 ^σ^
CATALASE	0.114 ± 0.040	0.113 (0.089–0.136)	0.082 ± 0.032	0.089 (0.057–0.098)	0.068 ^θ^
FRAP	119.0 ± 37.06	116.0 (86.13–148.5)	126.4 ± 17.08	125.2 (113.4–141.4)	0.597 ^θ^
CARBONYLPROTEIN	4.681 ± 3.293	4.088 (2.213–5.181)	3.309 ± 0.589	3.380 (2.865–3.650)	0.215 ^σ^

BC, Breast Cancer; BBD, Benign Breast Disease; GSH, Reduced Glutathione; SOD, Superoxide Dismutase; FRAP, Ferric Reducing Antioxidant Power; SD, standard deviation; ^θ^ t test and ^σ^ Mann Whitney.

**Table 4 cancers-14-05973-t004:** Profile of redox status markers among women with breast cancer HER2^+^, HER2^−^ and benign breast disease at a university hospital in Uberlandia, Minas Gerais, Brazil (*n* = 25).

Markers	HER2^+^ (*n* = 6)		HER2^−^ (*n* = 11)		BBD (*n* = 8)		*p*
	Mean ± SD	Median (p25–p75)	Mean ± SD	Median (p25–p75)	Mean ± SD	Median (p25–p75)	
GSH	1.317 ± 0.353	1.328 (0.998–1.607)	1.548 ± 0.701	1.335 (1.019–2.420)	2.021 ± 0.354	2.054 (1.831–2.195)	0.059 ^θ^
SOD	0.213 ± 0.021	0.210 (0.197–0.236)	0.203 ± 0.023	0.207 (0.175–0.219) ^a^	0.241 ± 0.023	0.230 (0.224–0.261) ^b^	0.010 ^σ^
CATALASE	0.096 ± 0.031	0.101 (0.065–0.120)	0.122 ± 0.042	0.122 (0.090–0.156)	0.082 ± 0.032	0.089 (0.057–0.098)	0.081 ^θ^
FRAP	94.59 ± 23.79 ^a^	87.63 (74.07–119.4)	132.3 ± 36.91 ^b^	131.3 (99.79–166.3)	126.4 ± 17.08	125.2 (113.4–141.4)	0.049 ^θ^
CARBONYL PROTEIN	5.629 ± 4.107	4.641 (3.208–7.322)	4.164 ± 2.844	3.991 (2.030–5.159)	3.309 ± 0.589	3.380 (2.865–3.650)	0.271^σ^

HER2+, human epidermal growth factor receptor 2 positive; HER2−, human epidermal growth factor receptor 2 negative; BBD, Benign Breast Disease; GSH, Reduced Glutathione; SOD, Superoxide Dismutase; FRAP, Ferric Reducing Antioxidant Power; SD, standard deviation; ^θ^ ANOVA test + Tukey’s multiple comparisons test; ^σ^ Kruskal-Wallis test + Dunn’s multiple comparisons test; ^a^ and ^b^, horizontal means/medians followed by different letters differed statistically according to the post-hoc test at the 5% probability level.

## Data Availability

The datasets used and/or analysed during the current study available from the corresponding author on reasonable request.

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
