# Peer review of "A Lower Serum Antioxidant Capacity as a Distinctive Feature for Women with HER2+ Breast Cancer: A Preliminary Study"

_cancers, 2022, doi:10.3390/cancers14235973_

Round 1
Reviewer 1 Report
This study investigates infrared spectrum wavenumbers obtained by ATR-FTIR and their relationship to the levels of redox status markers reduced glutathione, superoxide dismutase (SOD), catalase, Ferric Reducing Antioxidant Power (FRAP), and protein carbonyl among women with HER2+BC, HER2-BC and benign breast disease (BBD). The manuscript is well-written and publishable with minor revisions.
1. The manuscript should be updated with new articles from 2022.
2. Please, modify the style reference according to the style of the journal.
3. The resolution of the text inside Figure 2 is low and cannot be read.
4. Please mention the limitations of the study. Also, the limitations of using this method for early diagnosis should be mentioned.
5. The manuscript should be checked especially for spelling and grammatical errors. For example, superoxide dismutase is misspelled.
Author Response
Response to Reviewer (November 22, 2022)
We are grateful to the reviewer for the comments and suggestions. The requested modifications were very helpful for improving our manuscript “A lower serum antioxidant capacity as a distinctive feature for women with HER2+ breast cancer: a preliminary study”. Enclosed, please find the revised manuscript (all modifications have been highlighted in red). Below we provide the point-by-point responses.
Kind regards,
Yara Cristina de Paiva Maia, PhD
Molecular Biology and Nutrition Research Group
Federal University of Uberlandia, UFU.
Overall reviewer comments
[General Comment]: This study investigates infrared spectrum wavenumbers obtained by ATR-FTIR and their relationship to the levels of redox status markers reduced glutathione, superoxide dismutase (SOD), catalase, Ferric Reducing Antioxidant Power (FRAP), and protein carbonyl among women with HER2+ BC, HER2- BC and benign breast disease (BBD). The manuscript is well-written and publishable with minor revisions.
Response: Thank you very much for your comments and suggestions that helped us to improve our manuscript.
Specific Reviewer comments
[Comment 1] The manuscript should be updated with new articles from 2022.
Response: We thank you for the suggestion. We have updated the study with seven new articles from 2022.
[Comment 2] Please, modify the style reference according to the style of the journal.
Response: We appreciate the important note. We have updated the style reference according to the style of the journal.
[Comment 3] The resolution of the text inside Figure 2 is low and cannot be read.
Response: We thank the reviewer for this comment. We have improved the resolution of the Figure 2.
[Comment 4] Please mention the limitations of the study. Also, the limitations of using this method for early diagnosis should be mentioned.
Response: We thank the reviewer for this relevant suggestion. We have included two paragraphs to the end of the discussion (lines 389-398), bringing out the limitations:
“Although there are studies investigating the use of FTIR for cancer diagnosis, this was not the objective of the present study. Here we pioneered combining infrared spectrum analyzes and levels of redox status markers in the blood, expanding the frontier of scientific knowledge in this area for women with HER2+ BC, HER2- BC and BBD. We emphasize that, as a preliminary study, the results are reliable according to the a posteriori sample calculation, although the data presented are still from a small number of women. Another possible limitation is that, in addition to cancer, there are other comorbidities in the studied population. However, no differences were found for the values of the redox state markers when dichotomized into with and without comorbidities (results not shown).”
[Comment 5] The manuscript should be checked especially for spelling and grammatical errors. For example, superoxide dismutase is misspelled.
Response: Thank you very much for your comments. We double-checked the text and proofread it with a native English-speaking professional.

Reviewer 2 Report
The concept of work is extremely interesting. Especially since oxidoreductive disorders accompany many diseases. I will not refer to the methodology of the work, which was carefully selected. However, my doubts are raised by the comparison between the groups, which differ significantly in age, and the authors did not present the characteristics of the groups in terms of comorbidities. The groups are not comparable in terms of age, so the differences in oxidoreductive status may not only be due to cancer. In addition, the groups studied are small and heterogeneous in terms of age, disease progression and probably other factors such as smoking, medications used and dietary habits. The latter are very difficult to characterize in relation to oxidoreductive status.
Because of many uncertainties, especially clinical ones, I cannot qualify the paper for publication, and I would not want to unfairly judge the Authors. I propose to take into account my comments and improve the paper.
Author Response
Response to Reviewer (November 22, 2022)
We are grateful to the reviewer for the comments and suggestions. The requested modifications were very helpful for improving our manuscript “A lower serum antioxidant capacity as a distinctive feature for women with HER2+ breast cancer: a preliminary study”. Enclosed, please find the revised manuscript (all modifications have been highlighted in red).
Kind regards,
Yara Cristina de Paiva Maia, PhD
Molecular Biology and Nutrition Research Group
Federal University of Uberlandia, UFU.
Overall reviewer comments
[Comments and Suggestions for Author]: The concept of work is extremely interesting. Especially since oxidoreductive disorders accompany many diseases. I will not refer to the methodology of the work, which was carefully selected. However, my doubts are raised by the comparison between the groups, which differ significantly in age, and the authors did not present the characteristics of the groups in terms of comorbidities. The groups are not comparable in terms of age, so the differences in oxidoreductive status may not only be due to cancer. In addition, the groups studied are small and heterogeneous in terms of age, disease progression and probably other factors such as smoking, medications used and dietary habits. The latter are very difficult to characterize in relation to oxidoreductive status.
Because of many uncertainties, especially clinical ones, I cannot qualify the paper for publication, and I would not want to unfairly judge the Authors. I propose to take into account my comments and improve the paper.
Response: Thank you very much for your comments and suggestions that helped us to improve our manuscript.
Based on your question about comorbidities among the patients in our study, we carried out analyzes, verifying whether the diagnosis of other diseases brought a significant difference in the analyzed redox status markers. No differences were found when performing this sensitivity analysis.
Regarding age, we conducted sensitivity analyzes taking out the only two patients below 41 years old (one 19 years old BBD and one 29 years old HER2-) and found no differences in the direction of the results. Attached are tables 3 and 4 with the analyzes redone without these two participants. Thus, we would like to keep the analyzes as per the original tables in the article.
Breast cancer affects women of all age groups and with the most diverse comorbidities and lifestyle. We think for this reason that our study has a representative heterogeneity of participants, because in all groups there are women that exercise and women that don’t, smoke or don’t and consume alcoholic beverages or don’t. Furthermore, dichotomizing our sample for the variables mentioned above, no differences were found between groups for any redox status markers. So we think the differences in oxidoreductive status are due to the cancer and its worse or better prognosis.
We have experience in analyzing the total antioxidant capacity of the diet of women with breast cancer undergoing chemotherapy and we have an article published on this subject (doi: 10.3390/nu12113303), but this was not the objective of the current study, so we do not have these data for these volunteers. We have the perspective of increasing the sample size for a future study and inserting dietary data.
We add to the last paragraph of the discussion (lines 399-406): “Thus, our preliminary results are of great value to the scientific community as they indicate a distinctive feature of these HER2+ molecular subtypes, which should be extended to a larger sample population. More studies should be conducted with this innovative approach combining the ATR-FTIR technique and the evaluation of the redox status of women with BC.”
Finally, we would like to publish the present study with these preliminary results as a “short communication” in this prestigious journal aiming to expand the scientific frontier of breast cancer and the redox status.

Reviewer 3 Report
In this Communication the Authors report that the infrared spectrum wavenumbers obtained by ATR-FTIR can be related to the levels of blood redox status markers in women with breast cancer and benign breast disease. In overall, the presented data is original, interesting, well analyzed and gives highly valuable data for possible diagnosis of breast cancer prognosis. Based on the data presented, I would recommend minor revision, however, English language has to be corrected throughout the manuscript, so I recommend giving it to a native speaker or a professional!
Following are some of the language or general recommendations:
Throughout the manuscript decide whether it will be HER2+BC HER2-BC or HER2+ BC HER2- BC (with or without space).
Line 21-22: The sentence „These findings there is no precedent and allow us to assess biochemical changes that occur before clinical signs, monitoring the tumor.“ should be reformulated, in this way it does not grammatically make sense.
Line 27: correct “investigate” to “investigates”
Line 28 – a word is missing before “reduced”, for example “as”: “…and their relationship to the levels of redox status markers as reduced glutathione, superoxide dismutase (SOD), catalase,…“. Word “desmutase” also should be corrected!
Line 32: remove the word „that“ before „HER2+BC“
Line 40: word „monitoring“ should be replaced, for example by „indicating“
Line 81: add the word „to“ before „the standards of the Declaration…“
Line 86: write full words for abbreviations ER and PR
Line 103: who is „manufacturer“, can you add info about the company?
Line 123: correct „sorological“, and add „by“ before „ FRAP method“
Line 129: correct the sentence to „To detect protein carbonyls (nmol/mg ptn) in serum…“
Line 158: correct “high degree” to “high grade”
Line 163: correct “minimun” and “maximun” to “minimum” and “maximum”
Line 171-172: please write the combinations with a lower case “x”, because with the upper case it is confusing (HER2+XBBD etc); you can also add a space, as you did in the Table 2 column names. Please correct it throughout the manuscript.
Line 184-185: the text “T-test analysis of the second derivative spectra of 184 women with HER2+ BC (n = 6), HER2- BC (n = 11) and women with BBD (n = 8)” does not have to be under the table. The numbers of the participants can be added to the Title of the table
Lines 190-191: make superscript of “-1”
Figure 3: remove HER2+ x BBD above the graphs
Figure 4: (B) SOD and FRAP – the significance marks should be adjusted. SOD is significant for Her2- vs BBD, and FRAP for HER2- vs HER2+, but also for HER2+ vs BBD? This should be marked clearly and explained in the Figure legend which difference is significant.
Line 241: correct “affect” to “affects”
The discussion is quite long and has many parts which are not connected well. In the following text, I recommended removing some parts of it from the manuscript or removing some paragraphs to the other place in the discussion.
Line 266: correct to “Our results indicate HER2+ BC cases to be distinguished from HER2- BC and BBD…”
Line 267: rewrite “HER2+ BCXHER2- BC” here and in the other parts of the manuscript so it is clearer which group x which group, add some spaces, etc.
Line 289: The space is missing after the “…serum albumin[25].”, before the next sentence
Lines 294 to 297 are unnecessary and can be excluded from the manuscript. Does not bring any important information for the manuscript and is not discussed.
Line 298: correct to “…evaluated serum from healthy and women with BC using…”
Line 311: correct to “…clinical value in regard to medicine and nutrition…”
Lines 313-315: write 2 and 4 by words, and eleven as a number – that´s how usually the numbers are written
Line 327 to 338: this whole part also seems unnecessary in this manuscript
Line 385 to 394: I would recommend moving this paragraph closer to the beginning of the discussion, for example before the paragraph “Oxidative stress can cause spectral changes due to alterations of intramolecular and intermolecular interactions of the biomolecules present in the blood.”
Author Response
Response to Reviewer (November 22, 2022)
We are grateful to the reviewer for the comments and suggestions. The requested modifications were very helpful for improving our manuscript “A lower serum antioxidant capacity as a distinctive feature for women with HER2+ breast cancer: a preliminary study”. Enclosed, please find the revised manuscript (all modifications have been highlighted in red). Below we provide the point-by-point responses.
Kind regards,
Yara Cristina de Paiva Maia, PhD
Molecular Biology and Nutrition Research Group
Federal University of Uberlandia, UFU.
Overall reviewer comments
[General Comment]: In this Communication the Authors report that the infrared spectrum wavenumbers obtained by ATR-FTIR can be related to the levels of blood redox status markers in women with breast cancer and benign breast disease. In overall, the presented data is original, interesting, well analyzed and gives highly valuable data for possible diagnosis of breast cancer prognosis. Based on the data presented, I would recommend minor revision, however, English language has to be corrected throughout the manuscript, so I recommend giving it to a native speaker or a professional!
Response: Thank you very much for your comments and suggestions that helped us to improve our manuscript.
Specific Reviewer comments
[Comment 1] Throughout the manuscript decide whether it will be HER2+BC HER2-BC or HER2+ BC HER2- BC (with or without space).
Response: We thank you for the suggestion. We standardized all as HER2+ BC HER2- BC (with space).
[Comment 2] Line 21-22: The sentence „These findings there is no precedent and allow us to assess biochemical changes that occur before clinical signs, monitoring the tumor.“ should be reformulated, in this way it does not grammatically make sense.
Response: We appreciate the important suggestion and adjusted Lines 21-22 accordingly: “These unprecedent findings allowed us to assess biochemical changes that occur before clinical signs and allowed us to monitor the tumor.”
[Comment 3] Line 27: correct “investigate” to “investigates”.
Response: We thank the reviewer for this comment. We have made the change to the text and highlighted it in red.
[Comment 4] Line 28 – a word is missing before “reduced”, for example “as”: “…and their relationship to the levels of redox status markers as reduced glutathione, superoxide dismutase (SOD), catalase,…“. Word “desmutase” also should be corrected!
Response: We thank the reviewer for this comment. We have made the change to the text and highlighted it in red.
[Comment 5] Line 32: remove the word „that“ before „HER2+BC“.
Response: We thank the reviewer for this comment. We have made the change to the text and highlighted it in red.
[Comment 6] Line 40: word „monitoring“ should be replaced, for example by „indicating“.
Response: We thank the reviewer for this comment. We have made the change to the text and highlighted it in red.
[Comment 7] Line 81: add the word „to“ before „the standards of the Declaration…“.
Response: We thank the reviewer for this comment. We have made the change to the text and highlighted it in red.
[Comment 8] Line 86: write full words for abbreviations ER and PR.
Response: We thank the reviewer for this comment. We have made the change to the text and highlighted it in red: “estrogen receptor (ER), progesterone receptor (PR).”
[Comment 9] Line 103: who is „manufacturer“, can you add info about the company?
Response: We thank the reviewer for this comment. We have made the change to the text and highlighted it in red. The manufacturer is KASVI.
[Comment 10] Line 123: correct „sorological“, and add „by“ before „ FRAP method“.
Response: We thank the reviewer for this comment. We have made the change to the text and highlighted it in red.
[Comment 11] Line 129: correct the sentence to „To detect protein carbonyls (nmol/mg ptn) in serum…“.
Response: We thank the reviewer for this comment. We have made the change to the text and highlighted it in red.
[Comment 12] Line 158: correct “high degree” to “high grade”.
Response: We thank the reviewer for this comment. We have made the change to the text and highlighted it in red.
[Comment 13] Line 163: correct “minimun” and “maximun” to “minimum” and “maximum”.
Response: We thank the reviewer for this comment. We have made the change to the text and highlighted it in red.
[Comment 14] Line 171-172: please write the combinations with a lower case “x”, because with the upper case it is confusing (HER2+XBBD etc); you can also add a space, as you did in the Table 2 column names. Please correct it throughout the manuscript.
Response: We thank the reviewer for this comment. We have made the change to the text and highlighted it in red (HER2+ vs HER2-).
[Comment 15] Line 184-185: the text “T-test analysis of the second derivative spectra of 184 women with HER2+ BC (n = 6), HER2- BC (n = 11) and women with BBD (n = 8)” does not have to be under the table. The numbers of the participants can be added to the Title of the table.
Response: We thank the reviewer for this comment. We have made the change to the text and highlighted it in red.
[Comment 16] Lines 190-191: make superscript of “-1”.
Response: Thank you very much for your comments. We have made the change to the text and highlighted it in red.
[Comment 17] Figure 3: remove HER2+ x BBD above the graphs.
Response: Thank you very much for your comments. We removed HER2+ x BBD above the graphs.
[Comment 18] Figure 4: (B) SOD and FRAP – the significance marks should be adjusted. SOD is significant for Her2- vs BBD, and FRAP for HER2- vs HER2+, but also for HER2+ vs BBD? This should be marked clearly and explained in the Figure legend which difference is significant.
Response: Thank you very much for your comments. We adjusted and now the Figure 4 is correct.
[Comment 19] Line 241: correct “affect” to “affects”.
Response: We thank the reviewer for this comment. We have made the change to the text and highlighted it in red.
[Comment 20] The discussion is quite long and has many parts which are not connected well. In the following text, I recommended removing some parts of it from the manuscript or removing some paragraphs to the other place in the discussion.
Response: Thank you very much for your comments.
[Comment 21] Line 266: correct to “Our results indicate HER2+ BC cases to be distinguished from HER2- BC and BBD…”.
Response: We thank the reviewer for this comment. We have made the change to the text and highlighted it in red.
[Comment 22] Line 267: rewrite “HER2+ BCXHER2- BC” here and in the other parts of the manuscript so it is clearer which group x which group, add some spaces, etc.
Response: We thank the reviewer for this comment. We have made the change to the text and highlighted it in red.
[Comment 23] Line 289: The space is missing after the “…serum albumin[25].”, before the next sentence.
Response: Thank you very much for your comments. We have made the change to the text.
[Comment 24] Lines 294 to 297 are unnecessary and can be excluded from the manuscript. Does not bring any important information for the manuscript and is not discussed.
Response: Thank you very much for your comments. We have removed the mentioned paragraph.
[Comment 25] Line 298: correct to “…evaluated serum from healthy and women with BC using…”.
Response: We thank the reviewer for this comment. We have made the change to the text and highlighted it in red.
[Comment 26] Line 311: correct to “…clinical value in regard to medicine and nutrition…”.
Response: We thank the reviewer for this comment. We have made the change to the text and highlighted it in red.
[Comment 27] Lines 313-315: write 2 and 4 by words, and eleven as a number – that´s how usually the numbers are written.
Response: We thank the reviewer for this comment. We have made the change to the text and highlighted it in red: “Among the 17 women with BC who participated in the present study, only two (11.76%) were classified as G1 (well-differentiated tumor), 11 (64.70%) as G2 (moderately differentiated tumor) and four (23.52%) as G3 (poorly differentiated tumor).”
[Comment 28] Line 327 to 338: this whole part also seems unnecessary in this manuscript.
Response: Thank you very much for your comments. We rewrite the paragraph as follows and highlight it in red: “It is well known that diet is a key modifiable risk factor for several diseases, including BC. In the present study design, we did not collect data on the volunteers' diet, but our research group previously published a prospective study carried out among women with BC undergoing chemotherapy, where we found a decrease in the total antioxidant capacity of their diets over the time of treatment and among women that presented post-treatment metastasis.”
[Comment 29] Line 385 to 394: I would recommend moving this paragraph closer to the beginning of the discussion, for example before the paragraph “Oxidative stress can cause spectral changes due to alterations of intramolecular and intermolecular interactions of the biomolecules present in the blood.”
Response: We thank the reviewer for this comment. We have made the change to the text and highlighted it in red.

Round 2
Reviewer 2 Report
Thank you very much for the response to my review and the additions made by the Authors. I believe that the work will be more valuable once the study groups are increased, as the Authors announce in their response.
"We have the perspective of increasing the sample size for a future study and inserting dietary data."